# The Future of Targeted Treatment of Primary Sjögren’s Syndrome: A Focus on Extra-Glandular Pathology

**DOI:** 10.3390/ijms232214135

**Published:** 2022-11-16

**Authors:** Weizhen Zeng, Xinyao Zhou, Sulan Yu, Ruihua Liu, Chrystie Wan Ning Quek, Haozhe Yu, Ryan Yong Kiat Tay, Xiang Lin, Yun Feng

**Affiliations:** 1Department of Ophthalmology, Peking University Third Hospital, Beijing 100191, China; 2Department of Rheumatology, Guang’anmen Hospital, China Academy of Chinese Medical Sciences, Beijng 100053, China; 3School of Chinese Medicine, The University of Hong Kong, Hong Kong SAR, China; 4Yong Loo Lin School of Medicine, National University of Singapore, Singapore 119077, Singapore; 5State Key Laboratory of Pharmaceutical Biotechnology, The University of Hong Kong, Hong Kong SAR, China

**Keywords:** primary Sjögren’s syndrome, dryness, fatigue, depression, anxiety, clinical trials

## Abstract

Primary Sjögren’s syndrome (pSS) is a chronic, systemic autoimmune disease defined by exocrine gland hypofunction resulting in dry eyes and dry mouth. Despite increasing interest in biological therapies for pSS, achieving FDA-approval has been challenging due to numerous complications in the trials. The current literature lacks insight into a molecular-target-based approach to the development of biological therapies. This review focuses on novel research in newly defined drug targets and the latest clinical trials for pSS treatment. A literature search was conducted on ClinicalTrials.gov using the search term “Primary Sjögren’s syndrome”. Articles published in English between 2000 and 2021 were included. Our findings revealed potential targets for pSS treatment in clinical trials and the most recent advances in understanding the molecular mechanisms underlying the pathogenesis of pSS. A prominent gap in current trials is in overlooking the treatment of extraglandular symptoms such as fatigue, depression, and anxiety, which are present in most patients with pSS. Based on dryness and these symptom-directed therapies, emerging biological agents targeting inflammatory cytokines, signal pathways, and immune reaction have been studied and their efficacy and safety have been proven. Novel therapies may complement existing non-pharmacological methods of alleviating symptoms of pSS. Better grading systems that add extraglandular symptoms to gauge disease activity and severity should be created. The future of pSS therapies may lie in gene, stem-cell, and tissue-engineering therapies.

## 1. Introduction

Sjögren’s syndrome is a chronic systemic autoimmune disorder characterized by the destruction and diminished function of exocrine glands, mainly salivary and lacrimal glands, resulting in dry eyes and dry mouth [1]. In primary or secondary form, the disease is characterized by the gradual infiltration of lymphocytic cells and atrophy of glandular and ductal cells. 

Primary Sjögren’s syndrome (pSS) has become increasingly common with a prevalence of 0.3%~3% among the public. Clinical manifestations of the disease are largely classified into exocrine gland and extraglandular disease features, which may be associated with widespread systemic complications involving the thyroid, lungs, kidneys, liver, and nervous system. A wide range of cognitive issues in pSS have also been reported, where persistent fatigue occurs in approximately 70% of pSS patients. The dryness and fatigue symptoms of pSS often impose detrimental effects on quality of life, as patients suffer from difficulties in eating, sleeping, and interacting with others [2]. 

Consensus on the cytokines and molecular entities involved in the initiation and maintenance of chronic immune activation in the disease remains difficult to elucidate [1,3]. Currently, the literature surrounding these therapies remains fragmented and no studies has been done to link the cytokines involved in pSS to the novel biological agents. Recent studies on novel drugs have shown promising results for the improvement of disease prognosis but unintended effects on the immune system, a variety of complications, and rising costs for patients [4]. In addition, ESSDAI (EULAR Sjögren’s syndrome disease activity index), which is a primary outcome measure in most trials, does not include certain common symptoms such as vaginal dryness (occurring in up to 64% of female patients) [5,6], fatigue (70% of patients) [7], and mental effects (over 30% of patients) [8].

Given the prevalence and impact of pSS symptoms on quality of life, it is crucial to holistically assess the efficacy of these novel drugs in pSS treatment. This review discusses the updates of molecular mechanisms underlying the extraglandular pathogenesis and associated potential targets for pSS treatment, with a focus on these outcomes and complications of recent clinical trials. We adopt a two-pronged approach that first explores the therapeutic targets in pSS followed by delving into the treatment of pSS in a symptom-centred manner.

## 2. Potential T and B Cell Targets for the Development of Treatment Options for pSS

T and B cells are central to the pathogenesis of pSS. Given that this has already been extensively reviewed in the literature [3,9,10,11], this section discusses updates on potential targets and provides a framework to investigate the potential targets for disease-modifying therapy (Figure 1, Table A1). 

### 2.1. CD4+ T Helper Cells 

CD4+ T helper (Th) cells infiltrate target organs and produce multiple cytokines in the peripheral blood, inducing B cell activation, maturation, and production of autoimmune antibodies [9]. Th cell lineages are classically categorized according to their cytokine profile as interferon (IFN)-γ-producing Th1 cells, interleukin-4 (IL-4)-producing Th2 cells, and IL-17A (IL-17)-producing Th17 cells [12]. However, given the current understanding, these T cell subsets are also capable of producing various or overlapping cytokines. To understand the potential targets for disease-modifying therapy, this review adopts a recently revised classification system that divides the CD4+ Th cells into three subsets according to helper functions and discusses their roles and therapeutic potential in the SS pathogenesis [13]. 

#### 2.1.1. Type 1 Th Cells 

Type 1 Th cells mainly activate and attract mononuclear phagocytes such as macrophages, monocytes, and dendritic cells (DCs). These cells are primary executors of Type 1 immune responses for cellular immunity. Granulocyte–macrophage colony-stimulating factor (GM-CSF) and IFN-γ are the most canonical type 1 cytokines secreted by Type 1 Th cells [14]. 

GM-CSF serves as a powerful communication conduit between T cells and myeloid cells, promoting the survival of mononuclear phagocytes [14]. A research study using a murine model of dry-eye disease (DED) discovered that GM-CSF played a pertinent role in DED pathogenesis, not only limited to SS complication [10]. GM-CSF was found to be upregulated in the ocular surface, and GM-CSF-producing Th17 cells, a subset crucial for SS development, were found in greater abundance in draining lymph nodes [15,16]. Although being a non-canonical cytokine in classic Th cell lineages, GM-CSF diminished Th17 cell induction and caused an attenuation of corneal inflammation and a decrease in clinical signs of DED [15]. 

Despite its pivotal role in DED pathogenesis, therapeutic options targeting GM-CSF have yet to be realized in pSS treatment. Otilimab, a recombinant IgG1 monoclonal antibody against GM-CSF, has been developed for treating rheumatoid arthritis (RA) patients [17], while study in SS patients is not available. Therefore, further investigation into the use of GM-CSF-targeted biologics in pSS should be conducted to consider more potential role in the treatment of DED. 

IFN-γ is a canonical cytokine which induces the activation of T and NK cells and possesses dual roles in the immunomodulation [18]. Given the association between IFN-γ levels and lymphocyte infiltration in the exocrine glands, therapies targeted at IFN-γ should be considered [19].

#### 2.1.2. Type 2 Th Cells 

Type 2 Th cells target B cells and polymorphonuclear granulocytes like basophils, eosinophils, and mast cells. They generate a variety of cytokines such as IL-4, IL-21, and IL-25, which are mainly responsible for Type 2 responses in humoral immunity.

A cross-sectional study examining the labial salivary gland specimens of pSS patients found that specimens with GCs were nearly the only ones to exhibit IL-4-producing Th cells [20]. These cells promote B cell maturation and antibody production in the labial salivary glands, which highlights mechanisms causing xerostomia in pSS [20]. Another laboratory study of mice with absent or disrupted IL-4 genes, henceforth IL-4 knockout (KO) mice, discovered that IL-4 was necessary for disease progression from preclinical stage to clinical xerostomia [21]. Despite scarce data on IL-4, elimination or inhibition of IL-4 may be considered for prevention of severe clinical symptoms and signs in genetically predisposed patients. 

While IL-21 has not yet been considered as a target for pSS therapy, its role in germinal center (GC) formation, B cell activation, and differentiation to plasma cells in pSS makes it a suitable candidate for research [9,20,22]. A study of mice with experimental SS (ESS), patients with pSS, and patients with pSS-associated lymphoma revealed the critical role of IL-25 in pSS development [23]. In essence, the IL-25/IL-17RB axis was activated and both Th cells and IL-25 cytokines were detected in salivary epithelium particularly at the sites of chronic inflammation of patients and mice with pSS [23], making IL-25 a promising target for pSS therapy.

#### 2.1.3. Type 3 Th Cells 

Type 3 Th cells act on non-hematopoietic cells at barrier tissue sites comprising of epithelial cells and stromal cells [22]. They produce cytokines IL-17, IL-22, and IL-23, which exert Type 3 immune responses, mainly in barrier immunity. In vivo, significant increases in protein and mRNA levels of IL-22, IL-23, and IL-17 were detected in the inflamed salivary glands of pSS patients [24]. 

IL-17 serves important functions both in infectious and autoimmune diseases [16,25], and pSS is no exception. In a recent study, IL-17 KO mice did not express overt clinical pSS symptoms or histopathological features after induction of ESS compared with wild-type mice. Interestingly, after adoptive transfer of Th17 cells, this was restored [26]. This was corroborated by another study using bortezomib, a proteasome inhibitor that suppressed the secretion of IL-17, effectively ameliorated disease [27]. In addition, γδ T cell levels, which may be an indicator of IL-17 levels, were higher in the blood and salivary glands of pSS patients [10,28]. Evidently, IL-17 is required for the initiation and progression of pSS, while inhibition of IL-17 alleviates the disease. However, pSS patients are at a higher predisposition for oral candidiasis, and inhibition of IL-17 may increase the risk of fungal infections and other side effects, considering the antifungal properties of IL-17 [29].

IL-22 belongs to the IL-10 family of cytokines and acts by modulating tissue responses during inflammation [30]. In pSS, the effect of IL-22 in modulation of epithelial barrier remains to be elucidated [31]. A previous study demonstrated the role of IL-22 in inducing tertiary lymphoid-organ-induced pathology by promoting chemokine production in the epithelium, such as CXCL12 and CXCL13. Thus, this study provided evidence for using IL-22 blocking agents in B cell-mediated autoimmune diseases, which was also supported by a translational program of anti-IL-22 therapy in pSS conducted at the University of Birmingham [32]. 

The cytokine IL-23, which is mainly produced by antigen-presenting cells (APCs), is responsible for initiating the formation of IL-17-expressing Th cells as well as promoting the subsequent survival and effector functions of the Th cells and generating “pathogenic” Th17 cells [33]. Activation of the IL-23/IL-17 axis contributes to many inflammatory diseases such as pSS [34], making IL-23 as a promising therapeutic target for the production of disease-modifying drugs. Monoclonal antibodies against IL-23 in current clinical trials include ustekinumab, briakinumab, tildrakizumab, risankizumab, and guselkumab [35].

### 2.2. CD8+ Cytotoxic T Cells

CD4+ Th cells were traditionally thought to be the predominant T cell subtype infiltrating the glands of pSS. However, emerging evidence suggests CD8+ cytotoxic T cells may also play a role in pSS pathogenesis, whose chemotactic migration to target organs is driven by the activation of chemokine receptors. For instance, CXCR3 is upregulated upon chronic antigen stimulation in pSS [36]. However, considering the vital role of the chemotactic migration of CD8+ T cells in antiviral and anticancer mechanisms, targeting these chemokine receptors may not be appropriate in the treatment of pSS [37]. The essential roles of these cells are underscored by the attenuation of SS in mice with deficiency in CD8 [38]. Intraepithelial localization of CD8+ tissue-resident memory cells is mediated by interaction between integrin CD103 and E-cadherin in epithelial cells [39]. Thus, CD8+ tissue-resident memory cells are an important mediator in causing tissue damage in pSS and are a potential target for therapeutics.

### 2.3. Other Effectors Related with T Cells 

Some cytokines that are not exclusively produced by T cells, such as IL-1β and tumor necrosis factor α (TNF-α), were found increasing in pSS, hinting at their possible involvement in the mechanisms causing pSS [40].

#### 2.3.1. IL-1β 

IL-1β is closely involved in the pathogenesis of various autoimmune diseases as reviewed by Lopalco et al. [41]. Given the proteolytic activity of IL-1β, recent studies suggest that it may contribute to structural destruction of salivary glands in pSS patients [42]. 

#### 2.3.2. TNF-α 

The cytokine TNF-α is postulated to promote apoptosis of glandular epithelial cells and cause inflammation in salivary glands, while anti-TNF-α may protect the acinar structure and glandular function [43,44]. In salivary-gland-specific overexpression of TNF-α in a mouse model, severe inflammation and disruption of integrity of tight junction was observed [44]. Infliximab (anti-TNF-α) has proven to be essential in restoration of the proper localization of aquaporin-5, a water channel protein responsible for saliva and tear secretion. Moreover, TNF-α-induced matrix metallopeptidase 9 caused damages on extracellular matrix in exocrine glands, which could be prevented by anti-TNF-α agent [45]. Anti-TNF agents are now used in pSS [46] but have not received positive results in other studies due to the diverse effects of TNF-α on immune cell subsets [47].

#### 2.3.3. IL-36α and γδ T Cells 

IL-36α is a novel member of IL-1 family, deriving from αβ+ CD3+ T cells, CD68+ cells, and γδ T cells, where higher expression of IL-36α and IL-36R were detected in γδ T cells. IL-36α may be a driver of the DC maturation and Th1/Th17 responses in CD4+ T cells [48]. In pSS patients, the level of IL-36α increased in both the serum and salivary glands [49]. Given the role of γδ T cells in producing both IL-36 and IL-17, it was speculated that IL-17+ IL-36+ γδ T cells may participate in the pSS pathogenesis [49]. Given the pathogenicity of the effector T cells, a combinational therapy targeting anti-IL-36 together with first-line medication may potentiate therapeutic effects. 

#### 2.3.4. Treg Cells and IL-2 

Other players in the pathogenesis of pSS include regulatory T (Treg) cells in immune tolerance and IL-2 for growth and survival of Treg cells [50]. Low-dose IL-2 therapy showed promising disease outcomes in both murine and human autoimmunity and also restored the Treg/Th17 balance in pSS, but no obvious improvement was observed in short-term [51]. Considering the serious side effects caused by high-dose IL-2 treatment [52], long-term therapy using low-dose IL-2 may require additional evaluation in pSS. 

### 2.4. B Cells

The risk of developing B cell malignancy was highest in patients with pSS compared with other autoimmune diseases. In addition to antigen presentation, autoantibodies, and cytokine production, B cells also participate in tissue destruction [3,11]. Hence, this section focuses on the key molecules involved in B cell fate decision and function (Figure 2).

#### 2.4.1. CD40

The CD40 and CD40 ligand (CD40L or CD154) axis are central in T-B interactions in adaptive immunity [53], and targeting CD40 as therapy for pSS has been reviewed [54]. In an SS mouse model, the administration of MR1 (anti-CD154 blocking antibody) treatment suppressed ectopic lymphoid structures and inhibited salivary gland pathology [55]. Single injection of anti-CD40L in young mice prevented the formation of tertiary follicle neogenesis and production of autoantibodies involved in glandular pathology [56]. Similarly, iscalimab, an anti-CD40 monoclonal antibody, showed preliminary efficacy in treating pSS in a phase II clinical trial [57]. Therefore, targeting on CD40 may be a promising and feasible treatment for pSS. 

#### 2.4.2. CD20

CD20 is a phosphoprotein consisting of 44 amino acids. CD20 is expressed during the transition from pre-B cells to mature B cells and is lost during the development of plasma cells [58]. Rituximab, a monoclonal antibody against CD20, is widely used in many autoimmune conditions with different effect, making its use in pSS controversial. One clinical trial of 133 patients reported statistically significant (*p* < 0.05) improvement in unstimulated salivary flow rates at week 36 and 48 in the rituximab (anti-CD20) group (*n*  =  67) compared with placebo group (*n*  =  66) [59]. This was validated by the statistically significant increases in salivary flow rates in the group given rituximab at days 1 and 15 versus the placebo after 48 weeks [60]. However, its safety and long-term efficacy in pSS has yet to be shown, which may be attributed to the persistence of pathogenic autoantibodies secreted by plasma cells [61]. Anti-CD20 and anti-CatS/MHCII therapies may be viable options for the treatment of pSS symptoms.

#### 2.4.3. B Cell Activating Factor (BAFF)

BAFF, a member of the TNF superfamily proteins, is vital in promoting B cell survival and differentiation. It is found in high levels in serum and target organs of pSS, and thus, many therapeutic agents against BAFF have been created [62]. For example, sanalumab (VAY736), a BAFF-R inhibitor, induces antibody-dependent cellular cytotoxicity in B cells, which has shown to be useful in improving salivary function in pSS (NCT02149420). Belimumab, a fully human IgG1λ recombinant monoclonal antibody directed against BAFF, was used in a trial to treat pSS, resulting in 60% of patients reaching the primary outcome (improved ESSDAI and ESSPRI), although the dryness symptoms (salivary flow rate and Schirmer test) were not significantly changed [63]. Tibulizumab (LY3090106) (NCT04563195), a bispecific dual-antagonist antibody against BAFF and IL-17, influences both innate and adaptive immunity [64]. Lastly, iguratimod (IGU) inhibits nuclear factor-kappa B (NF-κB) activity and was found to inhibit BAFF level in peripheral blood of pSS (NCT04981145, NCT04830644) [65]. Given the efficacies of anti-B cell and anti-BAFF therapies in pSS treatment, sthe ynergistic effect of combining the two should be considered, and clinical trials of the combination of rituximab with belimumab to treat pSS are on-going [66].

#### 2.4.4. IL-6 

IL-6 is a pleiotropic cytokine with pro-inflammatory anti-inflammatory effects [67], whose secretion is critical for B-cell-driven T-cell-mediated autoimmune diseases. An experimental autoimmune encephalomyelitis murine model showed that mice deficient in IL-6-producing B-cells had milder disease than wild-type B-cell mice [68]. However, tocilizumab, an IL-6R inhibitor, did not affect pSS symptoms more than placebo in one clinical trial [69]. Also, given T-B cell interactions in spleen and lymph nodes, it is difficult to evaluate the efficacy of IL-6R inhibition at the immune synapse and its impact. Thus, while eliminating IL-6–producing B cells and targeting IL-6 might be a new direction for autoimmune disease therapy, it remains challenging. 

#### 2.4.5. IL-10 

IL-10-producing regulatory B (Breg) cells maintain immune tolerance in inflammatory and autoimmune pathologies. Impairment of Breg cell function with decreased IL-10 production led to increased disease activity in pSS patients and ESS mice [70]. Given the diverse phenotypes of Breg cells and the complexity of transcription factors involved in IL-10 transcription, targeting Breg cells in vivo remains an obstacle (Table A1).

## 3. Treatment of Exocrine Gland Disease in pSS

To fill in the lacuna of knowledge on the effect of novel therapies for the treatment of certain symptoms in pSS, a symptom-based approach is adopted in Section 4, Section 5 and Section 6. 

### 3.1. Dry Mouth (Xerostomia)

Salivary hypofunction leads to xerostomia in pSS patients. Dysphonia, dysphagia, stomatopyrosis (burning mouth), dysgeusia (altered taste), tooth erosion, and oral infections are common problems caused by xerostomia [60]. Current treatments include oral swabs, lip moisturizers, topical saliva substitutes, muscarinic agonists, scheduled use of ice water, electrostimulation, and acupuncture [71,72]. Though not life-threatening, dysphagia and awakening from sleep due to oral dryness are debilitating consequences of xerostomia [73], which beg the advancement of drug therapies beyond symptom relief [74]. This section reviews the possible molecular targets that should be investigated. 

Although type I IFN is a pro-inflammatory cytokine, low-dose IFNα treatment was found to increase aquaporin-5 transcription and protein production in human parotid gland tissue, resulting in enhanced saliva and tear secretion. Administration of 150 IU of oral IFNα three times daily demonstrated the greatest potential in improving salivary output in one study [75]. However, multiple adverse effects were discovered, including a flu-like syndrome, chest pain and arthropathy, central nervous system depression, and myelosuppression [75,76]. Alternatively, 10–15 mg weekly dose of methotrexate resulted in increased salivary flow rate [77]. Eculizumab, a humanized mAb binding to complement protein C5, has been shown to ameliorate fatigue in myasthenia gravis and may be considered for the alleviation of fatigue in pSS [78].

### 3.2. Dry Eye

Dry-eye disease (DED) manifests as itching, grittiness, irritation, foreign body sensation, and blurry vision in pSS patients [79]. Anti-inflammatory therapies for pSS-DED include DMARDs (disease-modifying anti-rheumatic drugs) such as iguratimod and methotrexate, topical corticosteroids (1% methylprednisolone), antibiotics (azithromycin, doxycycline, and minocycline), and immunosuppressive agents (cyclosporine A and tacrolimus) [80]. The variety of systemic and topical drugs and the intricate titration of doses pose a challenge to developing treatment for DED. Mediators in driving the lacrimal pathology, and the latest drugs being trialed for the relief of pSS-DED are discussed. 

Varieties of drugs targeting T cells have been considered in DED treatment. Abacept (NCT02067910, NCT04186871, NCT02915159) is a selective T cell costimulation inhibitor that may improving tear secretion in pSS by engaging CD28 and suppressing antigen-presenting cells. Baminercept, a lymphotoxin β receptor IgG fusion protein, inhibits differentiation and proliferation of T cells and decreases CXCL13 levels, which is associated with ectopic lymphocytic structures in the lacrimal and salivary glands of pSS patients [81]. However, the phase II trial failed to significantly improve either glandular or extraglandular pathology in pSS [82]. The combination of lulizumab (an anti-CD28 domain antibody antagonist) and BMS-986142 (a highly selective BTK inhibitor) is being trialed. Lastly, while hydroxychloroquine (HCQ) was recommended by the European League Against Rheumatism (EULAR) as an effective suppressor of effector T cells, it did not improve tear film break-up time and ocular surface disease index in a recent trial [83]. This was further supported by a meta-analysis, where the efficacy and efficiency of HCQ in relieving eye dryness was limited [84].

Other candidates for DED treatment are drugs targeting cysteine protease cathepsin S (CatS). CatS may play a crucial role in MHCII processing and T cell stimulation, as it was found to be elevated in the tears of pSS patients. RO5459072, a CatS inhibitor, caused a dose-dependent downregulation of CatS/MHCII-mediated effect and may be a potential target for treatment [85]. 

There are also emerging biological agents targeting Th1/Th17 cytokines and BAFF pathways. DMARDs such as leflunomide and anti-TNF, which ameliorate dryness symptoms in RA patients, may also be considered in pSS patients [86,87]. Topical use is preferred due to the side effects of these systemic immunosuppressants. Finally, 0.005%/0.01% lacripep and 0.05% cyclosporine eyedrops have also exhibited promising efficacy in relieving symptoms in pSS [88]. 

### 3.3. Vaginal Dryness and Dyspareunia 

Women with pSS may suffer from vulvovaginal dryness, vulvar pruritus, and dyspareunia, partially similar to that in postmenopausal women. Histopathological assessment of vaginal mucosa biopsies displayed subepithelial inflammation, with lymphocytic infiltrations of CD45+, CD3+, and B cells occurring more frequently in pSS patients. It has been postulated that this is induced by IFN-mediated CXCL10 and/or JAK-STAT pathways [6]. Lymphocytic infiltration resulting in decreased transudation of serous fluid into the vaginal vault has also been hypothesized. Clinical trials of interest are those on parsaclisib and abatacept. The results are highly anticipated as possible future therapies for vaginal symptoms in pSS.

### 3.4. Fatigue in pSS

Current trials often neglect the treatment of extraglandular symptoms such as fatigue, depression, and anxiety. Approximately 70% of pSS patients claim persistent fatigue [89,90]. Evidently, fatigue has significant effects on quality of life, rendering it as one of the key issues in pSS clinical management. Despite being the most common symptom in pSS, the immunology behind fatigue has yet to be established. Currently, factors involved in mechanisms causing fatigue in pSS include IL-1, IL-36α, and humoral autoimmunity. 

#### 3.4.1. IL-1 

IL-1 is a pro-inflammatory cytokine that exists in two biologically active forms: IL-1α and IL-1β [91,92]. Pharmacological experiments have shown that systemic administration of IL-1β to rats and mice induced a reduction in exercise activity, less food and water intake, social withdrawal, increase in slow wave sleep, and cognitive changes in a dose- and time-dependent manner that parallels human fatigue [93]. Studies have found that increased levels of IL-1 receptor antagonist (IL-1Ra) in the cerebrospinal fluid of pSS patients was associated with greater fatigue [94]. This was further supported by a cohort study (*n* = 49) that showed a higher association with serum IL-1β and hypocretin-1. Interestingly, hypocretin-1, the main regulator of sleep and wakefulness, is possibly driven by the IL-6/TNF-α axis, thus causing fatigue in pSS [95]. 

Treatment of fatigue in pSS has been challenging; however, inspiration may be drawn from a randomized study in which canakinumab, a human anti-IL-1β monoclonal antibody, improved the Short-Form Vitality 36 score from 12.0 to 48.3 in patients with gout [96]. Thus, targeting IL-1 may be an option for relieving fatigue in pSS.

#### 3.4.2. IL-36α 

As mentioned, IL-36α plays a role in pSS mechanisms. Interestingly, compared with pSS patients who did not experience fatigue, the expression of IL-36α was up-regulated in patients with fatigue [97]. Although there was no overt evidence for the function of IL-36α in causing fatigue in pSS, this possibility should be considered.

#### 3.4.3. Immunoglobulins 

It is speculated that the fatigue in pSS patients may be related to humoral autoimmunity. One study showed that the Fatigue Scale 14 in a sub-healthy population negatively correlated with their serum immunoglobulin A (IgA) and IgG levels. This was further corroborated by a cross-sectional study that demonstrated a positive correlation between increased IgG levels and risk of pSS-related fatigue [98]. 

#### 3.4.4. Other Mediators 

Recent studies found that the intensity of fatigue based on the Profile of Fatigue Questionnaire was negatively correlated with several pro-inflammatory cytokines, including IFN-γ, TNF-α, lymphotoxin α, and CXCL10 [99,100]. Analysis by logistic regression model revealed that lower levels of IFN-γ and CXCL10 with increases in reported pain and depression were the most important predictors of fatigue [99]. This constitutes an argument against the role of inflammation in the pathogenesis of fatigue in pSS; more research is needed given its prevalence and pervasiveness.

#### 3.4.5. Alternative Medicine 

The future of pSS therapy should consist of a blend of Western and alternative medicine, tapping their synergistic potential while maintaining a balance to minimize the risk of drug–drug interactions. Total glucosides of paeony (TGP), derived from the herb root of the Paeonia lactiflora pall, was approved by the Food and Drug Administration of China to enter the market as a DMARD since 1998. A multi-center study found that TGP improved fatigue VAS scores [101]. The mechanisms are based on the balancing of Th1/Th2 cytokines and reduction of IFN-γ, IL-4, Fas, and FasL expression, as revealed on serological assessment [102]. Acupuncture therapy is a well-recognized approach by the public for relieving fatigue. A protocol for a trial was published in 2017 [103], and ongoing study of acupuncture treatment in pSS may provide evidence for it ameliorating fatigue. 

#### 3.4.6. Anti-Inflammatory and Immunosuppressive Treatment for Fatigue 

The therapies targeting IL-1, IL-36α, and immunoglobulins for pSS-related fatigue and current treatments, efficacies, and adverse effects are as follows, mainly including hydroxychloroquine (HCQ), rituximab, and TNF-α inhibitors.

A retrospective study of sham-needle-free group in 1996 reported that about one-third of systemic lupus erythematosus (SLE) patients with SS treated with HCQ had an improvement in fatigue [104]. Based on this, HCQ has been considered for the treatment of fatigue symptoms in pSS. Although a randomized experiment completed in 2012 (NCT00632866) found that HCQ has limited efficacy in improving fatigue [105], the 2016 guidelines continue to support the use of HCQ in selected situations [106].

Rituximab is associated with better visual analogue scale (VAS) scores for fatigue in 17 pSS patients receiving 1000 mg rituximab for 6 months [107]. A larger trial (*n* = 120) had similar findings, where VAS scores for fatigue in patients treated with rituximab were also improved [108] due to the elimination of B cell-mediated immune response and immunoglobulin productions. 

A pilot study reported that fatigue symptoms improved in pSS patients treated with infliximab as an inhibitor against TNF-α signaling [109]. A hypothesis from an in vitro study suggested that, apart from the blockage of circulating TNF-α molecules, infliximab also has the added function of inhibiting membrane-bound TNF-α. This may explain why infliximab-mediated interruption has a longer duration of action compared with etanercept, which does not improve fatigue because the receptor fusion protein detaches from membrane-bound TNF-α [110]. Similarly, there is a lack of evidence on whether the latest biological drugs, such as anakinra (NCT00683345) [111], abatacept (2009-015558-40) [112], belimumab [63], and epalizumab [113] could be applied for treating fatigue in patients with pSS.

Additionally, dehydroepiandrosterone (DHEA) has been proposed as a treatment for several autoimmune diseases. However, a trial suggested that there were no significant differences in fatigue between pSS patients treated with DHEA or placebo [114]. These findings were supported by another study (NCT00543166) that demonstrated that DHEA substitution treatment in DHEA-deficient and severely fatigued patients with pSS did not significantly improve fatigue compared to the placebo [115]. 

### 3.5. Depression and Anxiety in pSS

Psychological complications frequently disturb patients with fatigue and pain [116,117]. Depression and anxiety occurs in 36.9% and 33.8% of pSS patients in China [118]. In France, anxiety (pSS: 41.5%, non-pSS:39.5%) and depression (pSS: 28.3%, non-pSS: 26.7%) happen more frequently in pSS patients [119]. A study found that patients who experienced more pain, fatigue, worse oral hygiene, and swallowing disorders also had greater anxiety and depression. In female pSS patients, pain (β = 0.025, *p* = 0.028) and fatigue levels (β = 0.029, *p* = 0.004) were associated with anxiety, while pain (β = 0.022, *p* = 0.047), fatigue (β = 0.033, *p* = 0.001), and xeroderma scores (β = 0.030, *p* = 0.003) were strongly associated with depression [117]. Meanwhile, oral health (OR =  0.956, *p* < 0.05) and swallowing disorders (OR = 1.036, *p* < 0.05) were significantly associated with anxiety, while fatigue (OR = 0.587, *p* < 0.05) was positively correlated with depression in pSS [8,117].

To date, the underlying mechanisms of depression or anxiety in pSS remain unclear, although neurobiological factors are speculated. A recent hypothesis suggests that depression may be attributed to neuronal serotonergic and noradrenergic dysfunction, the change of dopamine and brain-derived neurotrophic factor (BDNF), and hyperactivity of hypothalamic–pituitary–adrenal (HPA) axis in the central nervous system. Additionally, gut microbe and amino acid metabolism and autoAbs against neuropeptides have also been considered to be involved in the pathological mechanisms of depression [120]. Early studies reported relatively higher serum autoAbs against α-melanocyte-stimulating hormone (MSH) in pSS patients compared with health controls. This was highly correlated with anxiety states [121], which may result from dysregulation of melanocortin system. A study found an association between humoral autoimmunity and cytokines with depression [122]. While these findings were similar to those in animal models, greater information is needed about anxiety and depression in humans. 

Interestingly, BAFF transgenic mice were established to research the mechanisms of anxiety, supporting the notion that humoral autoimmunity may partially be responsible for brain inflammation, impaired neurogenesis (stress-related brain responses), and hippocampal plasticity, leading to pSS-related anxiety [123]. Notably, using the same model, further studies suggested that dietary supplementation with *n*-3 polyunsaturated fatty acids could inhibit hippocampal microglial activation and increase hippocampal progenitor cell proliferation and plasticity [124], thus validating the effect of humoral autoimmunity-mediated neuroinflammation on depression. Given the possible role of BAFF in depression, it is worthwhile to explore the BAFF-targeted drugs mentioned above (Table A2).

Except for regular drug options like serotonin selective reuptake inhibitors (SSRIs) [125], compelling evidence has suggested that traditional Chinese medicine can relieve anxiety and depression in pSS [126], including the influence of BAFF production and endocrine hormone levels. Additionally, in a randomized-controlled trial (*n* = 45) (NCT02370225), aerobic exercise greatly improved life quality by relieving fatigue and depression, further supported by studies showing supervised walking as beneficial for female pSS patients [127]. Additionally, acupuncture (NCT02691377) and dehydroepiandrosterone (NCT00391924) were also shown to have favorable effects on depression and anxiety, though the mechanisms remain unclear. Thus, a combination of neuroinflammation resolution and aerobic exercise may be considered as alternative approaches for treating pSS patients with depression or anxiety.

## 4. Discussion

At present, most trials investigating the use of biologics in pSS patients are in phase II. Using the available tools, ESSDAI and some questionnaire scores, the current status of therapeutic outcomes may be somewhat confusing due to the subjectivity of the data. Sometimes, the improvement in life quality for patients with pSS is far from satisfactory, and the number of approved biologics is relatively small. The challenge lies in developing therapies for a condition with unestablished and complicated mechanisms. Innovative and bolder methods should be explored.

Two such methods are gene and stem-cell treatment. One experiment revealed that the transfer of human aquaporin-1 gene to damaged salivary glands enhanced parotid salivary flow. This safe method in humans is currently an impractical solution to a simple problem but could expand the horizons of gene therapy, stem-cell, and tissue-engineering therapy in treatment development in pSS [128]. Success in organoids and rodent models has paved the way for translation into humans [129]. Research should be expanded in the area of stem-cell and tissue-engineering therapies, which may one day offer cures for pSS. We envision the use of these techniques to replace or treat the glands affected in pSS to halt the infiltration of lymphocytic cells and atrophy of glandular and ductal cells.

Given the overlapping pathogenesis and mechanisms in autoimmune diseases, such as RA and SLE, future research should consider therapies used in these conditions that may be translatable to pSS. Currently, 22 out of 140 such trials meet this criterion; examples include HCQ, IGU, and abatacept, which are drugs used in RA that are currently applied to pSS [130].

ESSDAI is a systemic disease activity index that was designed to measure disease activity in patients with pSS. Dry-eye symptoms were mostly disregarded in outcome measures in clinical trials of drugs targeting B cells. Despite the debilitating effects on quality of life and sexual ability, vaginal dryness, the most common gynecological involvement, is still often not considered as a primary outcome. A more innovative and creative direction should be taken in pSS treatment research, which might involve discussions with neuroscientists and neuropsychiatrists.

## 5. Methods

The data included in this narrative review were identified via a literature search of the electronic database ClinicalTrials.gov using the search term “Primary Sjögren’s syndrome”. Only English language articles published between 2000 and 2022 were included. The results were curated by the authors for their relevance in understanding the pathogenesis and treatment options of pSS.

## 6. Conclusions

In conclusion, this study reviewed the emerging therapies of pSS and analyzed them in conjunction with cytokines involved in pSS pathogenesis. Management of pSS should extend beyond symptomatic treatment and consider the mechanisms that underlie pSS to get to the root of the problem. It is critical that pSS treatment be multipronged, using a combination of non-pharmacological, pharmacological, synthetic, and biological therapy to optimize treatment by drawing out synergistic effects. The therapeutic potential of DMARDs used in other autoimmune conditions may be considered in pSS treatment in view of the parallels in their pathophysiology. Future research should focus on these pertinent points in any approach to developing therapies that may treat or improve the life quality of pSS patients.

## Figures and Tables

**Figure 1 ijms-23-14135-f001:**
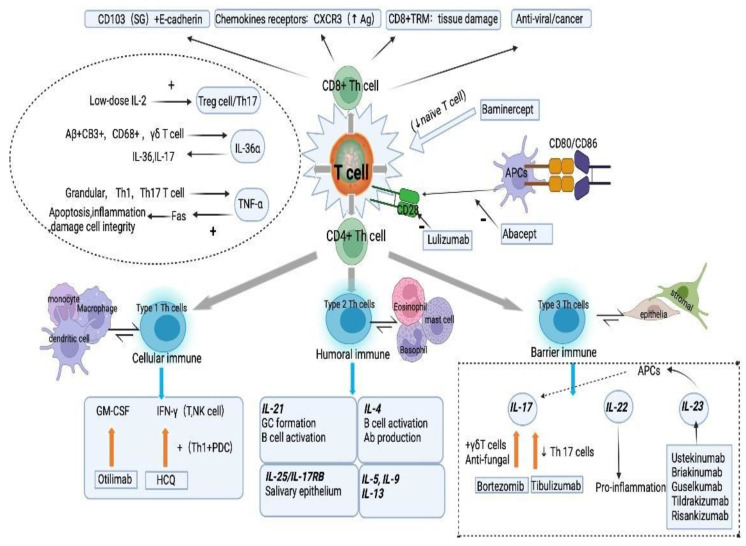
The effect of T cells on the pathologics of pSS.

**Figure 2 ijms-23-14135-f002:**
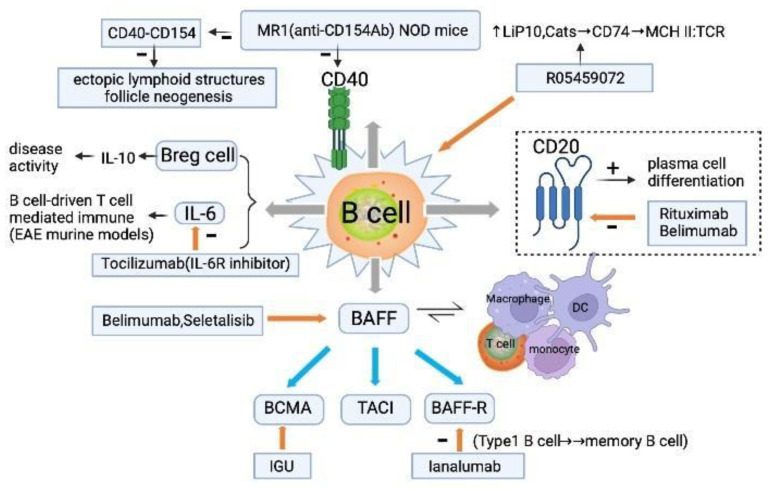
The effect of B cells in the pathologics of pSS.

## Data Availability

Not applicable.

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
