# Peer review of "The Future of Targeted Treatment of Primary Sjögren’s Syndrome: A Focus on Extra-Glandular Pathology"

_ijms, 2022, doi:10.3390/ijms232214135_

Round 1

Reviewer 1 Report

A summary table is recommend for section 2 (line 68)

Author Response

We highly appreciated the insightful comment from the reviewer. In line with the suggestion, we have added a new table to summarize the content for indexing in the revised manuscript. 

Reviewer 2 Report

This review paper revealed the potential targets for pSS treatment in clinical trials and the most recent advances in understanding of molecular mechanisms underlying the pathogenesis of pSS. Based on dryness and these symptom-directed therapies, emerging biological agents targeting inflammatory cytokines, signal pathways and immune reaction have been studied and proved the efficacy and safety. Novel therapies may complement existing non-pharmacological methods of alleviating symptoms of pSS. Better grading systems adding extraglandular symptoms to gauge disease activity and severity should be created. The future of pSS therapies may lie in gene therapy, stem cell and tissue engineering therapy.

The results, discussion, and conclusion of this paper were important in this topic. This paper could be accepted for publication.

Author Response

We highly appreciated the recognition by the reviewer. 

Reviewer 3 Report

nothing

Author Response

We thank for the reviewer's recognition and have made some changes on the content. Additional table has been added to help the readers in understanding and indexing the section 2.
